# Epidemiology of Glioblastoma Multiforme–Literature Review

**DOI:** 10.3390/cancers14102412

**Published:** 2022-05-13

**Authors:** Szymon Grochans, Anna Maria Cybulska, Donata Simińska, Jan Korbecki, Klaudyna Kojder, Dariusz Chlubek, Irena Baranowska-Bosiacka

**Affiliations:** 1Department of Biochemistry and Medical Chemistry, Pomeranian Medical University in Szczecin, Powstańców Wielkopolskich. 72 St., 70-111 Szczecin, Poland; szymongrochans@gmail.com (S.G.); d.siminska391@gmail.com (D.S.); jan.korbecki@onet.eu (J.K.); dchlubek@pum.edu.pl (D.C.); ika@pum.edu.pl (I.B.-B.); 2Department of Nursing, Pomeranian Medical University in Szczecin, Żołnierska 48 St., 71-210 Szczecin, Poland; 3Department of Ruminants Science, Faculty of Biotechnology and Animal Husbandry, West Pomeranian University of Technology, Klemensa Janickiego 29 St., 71-270 Szczecin, Poland; 4Department of Anaesthesiology and Intensive Care, Pomeranian Medical University in Szczecin, Unii Lubelskiej 1 St., 71-281 Szczecin, Poland; klaudyna.kojder@pum.edu.pl

**Keywords:** glioblastoma multiforme, epidemiology, risk factor, protective factor, brain tumor, central nervous system

## Abstract

**Simple Summary:**

Glioblastoma multiforme (GBM) is one of the most aggressive malignancies, accounting for 14.5% of all central nervous system tumors and 48.6% of malignant central nervous system tumors. The median overall survival (OS) of GBM patients is only 15 months. The aim of this review was to provide an overview of the epidemiology of GBM and factors that may have a significant impact on the risk of GBM.

**Abstract:**

Glioblastoma multiforme (GBM) is one of the most aggressive malignancies, with a median overall survival of approximately 15 months. In this review, we analyze the pathogenesis of GBM, as well as epidemiological data, by age, gender, and tumor location. The data indicate that GBM is the higher-grade primary brain tumor and is significantly more common in men. The risk of being diagnosed with glioma increases with age, and median survival remains low, despite medical advances. In addition, it is difficult to determine clearly how GBM is influenced by stimulants, certain medications (e.g., NSAIDs), cell phone use, and exposure to heavy metals.

## 1. Introduction

Glioblastoma multiforme (GBM) is one of the most aggressive malignancies and also the most common malignant primary tumor of the brain and central nervous system, accounting for 14.5% of all central nervous system tumors and 48.6% of malignant central nervous system tumors [1]. The median overall survival (OS) of GBM patients is low, at only 15 months [1,2].

GBM originates from astrocytic glial cells [3], and is the higher-grade malignant glioma (grade IV). It is difficult to clearly define the incidence of GBM as it varies depending on the report, from 3.19 cases per 100,000 person-years [4,5] to 4.17 per 100,000 person-years [6]. The incidence in the pediatric population (0–18 years) is 0.85 per 100,000, where pediatric glioblastoma multiforme (p-GBM) accounts for 3–15% of primary brain tumors [7,8,9,10,11,12] in this age group, although primary tumors of the central nervous system are the second-most-common type of cancer in children and the most common among solid tumors in children [13]. Ostrom et al. [14] presented an age-adjusted incidence rate of 0.18 (95% CI 0.16–0.19) per 100,000 people in the 0–19-year-old US population.

Due to the limited number of publications on the epidemiology of Glioblastoma multiforme, we have relied, in many respects, on works on glioma tumors, which include GBM. These studies cannot be put on par with studies describing GBM alone, but they allow us to approximate and build on existing knowledge potentially relevant to GBM.

## 2. Classification of GBM

The international standard for the nomenclature and diagnosis of gliomas is the World Health Organization (WHO) classification, in which GBM is a grade-IV cancer.

The fourth WHO classification of gliomas from 2016 is based on the degree of malignancy, as determined by histopathological criteria, in which four types of this neoplasm have been distinguished [15]:**Glioblastoma, *isocitrate dehydrogenase*** (***IDH*)** wildtype (90% of cases), developing de novo at about 60 years of age;**Glioblastoma, *IDH*-mutant** (10% of cases), secondary GBM that usually develops in younger patients with gliomas of higher differentiation (WHO grades I-III); it carries a significantly better prognosis than wildtype *IDH* [16];**Glioblastoma not otherwise specified (NOS)**, the *IDH* mutation status could not be determined due to a lack of histological or molecular material for testing;**Not-elsewhere-classified (NEC****) Glioblastoma**, fourth category distinguished in recent years. The necessary determinations regarding the classification of the tumor have been made, but the results do not allow matching the tumor to any of the aforementioned categories of the 2016 WHO division. This situation may occur in the case of discrepancies between the clinical, histological, immunohistological, and genetic features of the tumor. There is also the possibility that a GBM subunit exists with an unknown combination of features that is not yet classified in the WHO division.

It is worth noting that the 2016 classification of gliomas takes into account not only the histological classification of the tumor, but also the molecular changes within the cells [17].

The mutation of *IDH1* and *IDH2* reduces the affinity of enzyme isocitrate dehydrogenase for the standard substrate, isocitrate, and increases its affinity towards alpha-ketoglutarate. This in turn results in the conversion of alpha-ketoglutarate to the oncometabolite 2-hydroxyglutarate, a compound implicated in the inhibition of cell differentiation via a competitive blockade of α-ketoglutarate-dependent dioxygenases. This triggers the blockade of dioxygenase functions—DNA and histone demethylation [18]—and leads to the activation of hypoxia-inducible factor 1 (HIF-1) [19]. It is now recognized that these epigenetic changes are mainly involved in the formation of low-grade gliomas [18]. Further transformation of low-grade gliomas with an IDH mutation to secondary GBM requires more genetic alterations, such as the amplification of epidermal growth factor receptor (EGFR) and lost expression of the phosphatase and tensin homolog (PTEN) [20].

The trend of GBM genotyping is expanded in the new WHO classification. Increasingly widespread molecular profiling and the use of machine learning methods lead to the more accurate determination of prognoses and responses to specific treatments [21]. The discovery of new mutations present in GBM offers the potential for the creation of new drugs with new targets, and the association of mutations within tumors with different clinical courses will facilitate the diagnosis and prediction of disease severity. In 2017, the formation of cIMPACT-NOW (Consortium to Inform Molecular and Practical Approaches to central nervous system (CNS) Tumor Taxonomy) was announced to evaluate and recommend changes to the WHO classification of brain tumors [22].

The latest criteria and nomenclature introduced by the WHO in 2021 continue to strengthen the role of molecular genetics in the diagnosis of GBM. IDH-mutant tumors that could previously be classified as diffuse astrocytoma, anaplastic astrocytoma, or glioblastoma are currently considered a single type of IDH-mutant astrocytoma graded II, III, or IV. Moreover, grading IDH-mutant diffuse astrocytic tumors is no longer purely histological—it is also based on the presence of the CDKN2A/B homozygous deletion mutation, which results in a CNS WHO grade of IV, even in the absence of microvascular proliferation or necrosis [23].

The classification also includes TERT promoter mutation, EGFR gene amplification, and combined gain of entire chromosome 7 and loss of entire chromosome 10 (+7/−10) as qualifying for the diagnosis of GBM, IDH-wildtype. As a result, GBM, IDH-wildtype should be diagnosed in the case of an IDH-wildtype diffuse and astrocytic glioma in adults if there is either microvascular proliferation, necrosis, TERT promoter mutation, EGFR gene amplification, or +7/−10 chromosome copy number changes. Pediatric patients, however, should be diagnosed for IDH-wildtype diffuse astrocytomas using different types of pediatric-type gliomas [23].

## 3. Pathogenesis of GBM

### 3.1. GBM Site in the Brain

GBM is far more commonly localized in the supratentorial compartment [24] than the subtentorial compartment, and most frequently in the frontal lobe [25]. The rarest locations of GBM are the brainstem and cerebellum (Figure 1).

### 3.2. Genetic Pathogenesis

Modern advances in genomic technology have improved our understanding of the key molecular changes that trigger GBM. The molecular markers described below carry prognostic and predictive information, facilitate the differentiation of specific tumor types, and offer hope for understanding tumor progression and creating targeted therapies.

***ATRX* (*a-thalassemia/mental-retardation-syndrome-X-linked*)**
**mutation.** The *ATRX* gene located on Xq21.1 encodes a protein involved in the chromatin-rearrangement pathway, allowing histone H3.3 to be incorporated into heterochromatin [33]. *ATRX* mutations occur in approximately 57% of secondary GBM; in GBM cells, *ATRX* mutations occur more frequently in *IDH*-mutant GBM than in wildtype (71% vs. exceptions) [15] and co-occur with *IDH1* and *TP53* mutations [34,35]. *ATRX* mutations are positive prognostic factors [36,37]. In a prospective study conducted on a cohort of patients with astrocytic tumors (grade I-IV), those with a loss of ATRX expression had a better prognosis than those who retained ATRX expression and a co-occurring *IDH* mutation [38].***TERT* (*Telomerase Reverse Transcriptase*)** promoter mutation. The *TERT* gene encodes telomerase, an enzyme responsible for adding the missing 3′ end of a DNA strand during replication. The mutation of the *TERT* gene promoter results in increased telomerase activity and telomere elongation, suggesting that maintaining the presence of telomeres is essential for brain tumor formation [39,40]. The two most common mutations of the *TERT* promoter are C228T and C250T, located at base pairs 124 and 146, respectively, which encode this promoter [41]. These mutations can lead to up to a fourfold increase in *TERT* expression [39,42]. *TERT* mutations are present in up to 80% of GBM [39,43,44,45,46]. The role of *TERT* promoter mutation as a prognostic factor has not been unequivocally determined due to numerous confounding co-occurring factors (age, surgical intervention, *IDH*, *EGFR* mutations, and *MGMT* methylation status) [47]. *TERT* promoter mutations occur more frequently in *IDH*-wildtype GBM than *IDH*-mutant GBM (72% vs. 26%) [15]. Further prospective studies are needed on large cohorts of a homogeneous patient population (for example, *IDH*-wildtype and *O^6^-methylguanine DNA methyltransferase* (*MGMT)* promoter-unmethylated glioma) to independently assess the prognostic impact of *TERT* promoter mutations [47].***TP53* (*Tumor protein P53*)**
**mutation:** The *TP53* gene is located on human chromosome 17p13.1. The functional p53 protein is a homotetramer that plays a key role in the regulatory network, controlling proliferation, survival, genome integrity, and other cellular functions. The presence of *TP53* mutations is associated with the progression of GBM [48]. The inactivation of p53 correlates with increased invasiveness [49], decreased cell apoptosis [50], and increased proliferation [51]. Cell lines carrying the p53-inactivating mutation show greater resistance to DNA-damaging chemotherapeutics, such as cisplatin [50]. Although *TP53* mutations correlate with poor prognoses in other cancers [52], they have no prognostic value in GBM [51,53,54]. *TP53* mutations are more common in *IDH*-mutant GBM than *IDH*-wildtype GBM (81% vs. 27%) [15]. The gain of function (GOF) *TP53* mutation results in a new function or altered expression; in GBM, it leads to increased malignancy of cells by increasing their proliferation, migration, invasion, metastasis, drug resistance, and genome instability, and increasing survival [55,56,57,58]. Wang, Xiang et al. [59] reported that GOF mutations are associated with worse OS, and that they reduce GBM sensitivity to temozolomide by increasing *MGMT* expression [59].**B-RAF V600E mutation:** B-RAF is part of the RAS-RAF-MEK-ERK MAP kinase pathway. This precisely regulated pathway is responsible for cell growth; mutations that confer the constitutive B-RAF kinase activity would result in uncontrolled cell proliferation and tumor formation. The V600E mutation involves the substitution of valine for glutamate at position 600 of the B-RAF protein, producing the permanently activated serine/threonine kinase B-RAF, which activates extracellular signal-regulated kinase 1 and 2 (ERK1/2) and other mitogen-activated protein (MAP) kinases. In the literature, the frequency of all B-RAF mutations in GBM is estimated at 2–6%. In one study [60], patients with GBM from four studies were evaluated, and 8 out of 505 (1.5%) showed the presence of the B-RAF V600E mutation. The mutation may be a convenient point of entry for effective personalized anticancer therapy with kinase inhibitors, as evidenced by published case reports, such as the clinical response to vemurafenib (a B-RAF kinase inhibitor) in three pediatric high-grade gliomas [61].**GATA4** (GATA-binding protein 4)**:** GATA4 is a transcription factor of the GATA6 family, considered a suppressor gene. In normal astrocytes, GATA4 does not affect cell growth; however, in mice with knockout *GATA4* genes and null-p53 status, the absence of GATA4 induces a transformation associated with increased proliferation, resistance to chemotherapy, and radiotherapy-induced apoptosis [62]. A series of studies conducted by Agnihotri et al. [63] showed that: (i) in 94/163 human GBM tumor cells, GATA4 expression was lost, (ii) GATA4 inhibited transformation to GBM in vitro and in vivo, and (iii) the re-expression of GATA4 in GBM cells sensitized them to temozolomide, regardless of the MGMT mutation status. The role of GATA4 in response to temozolomide suggests the utility of the GATA4 mutation status as a predictive biomarker, and this needs to be confirmed in further studies.**FGFR1** (Fibroblast Growth Factor Receptor (1)): the FGFR family of proteins is a group of transmembrane receptors with tyrosine kinase function. The exact impact of signaling by FGFR on the pathobiological aspects of individual cancers remains unknown [64]. The strongest evidence suggests that FGFR1 contributes to poor prognosis in GBM, and signaling through this pathway is associated with increased radioresistance, invasiveness, and stemness [65,66,67].**EGFR** (epidermal growth factor receptor): EGFR is a receptor with tyrosine kinase activity that is activated by EGF (epidermal growth factor). EGFR promotes cell proliferation by activating the MAPK and PI3K-Akt pathways [68]. The EGFR gene is located at locus 7p12 and its amplification is observed in approximately 40% of GBM cases [69]. The amplification of EGFR has been associated with poor prognoses by some authors, but the results are inconclusive [70,71,72,73,74]. EGFR amplification is more common in *IDH*-wildtype GBM than *IDH*-mutant GBM (35% vs. exceptional) [15]. The most common EGFR mutation is variant-III EGFR mutation (EGFRvIII), involving a deletion without a shift of the reading frame of base pair 801 extending from exon 2 to 7, and it has attracted much research interest. This mutation leads to the constitutive activation of EGFR, resulting in the activation of downstream tyrosine kinase pathways. EGFRvIII mutation occurs almost exclusively in the presence of EGFR amplification [75]. Attempts to create a vaccine targeting EGFRvIII (Rindopepimut) ended in the phase-IV clinical trial in 2016 due to a lack of improvement in OS.**MGMT** (O^6^-methylguanine DNA methyltransferase): the *MGMT* gene is located on chromosome 10q26 and encodes a protein responsible for DNA repair, removing an alkyl group from the O^6^ position of guanine, an important DNA alkylation site. The presence of MGMT promoter methylation is a positive predictor of better OS. The authors of a study published in *NEJM* [76] suggested the usefulness of determining the *MGMT* promoter methylation status by methylation-specific PCR in order to identify patients who may benefit from including temozolomide with standard radiotherapy compared with radiotherapy alone. Temozolomide works by methylating DNA at the N^7^ and O^6^ atoms for guanine and N^3^ for adenine. Methylation sites can be repaired by specialized enzymes, such as MGMT. In a paper published in 2012 [77], the efficacy of temozolomide was greater for a methylated *MGMT* promoter in GBM cells (i.e., reduced MGMT expression). The use of the MGMT inhibitor O^6^-benzylguanine (O^6^-BG) restored temozolomide (TMZ) sensitivity to TMZ-resistant cell lines LN-18 and T98G [78].***WT1* (*****The Wilms tumor gene*****)**: *WT1* was first identified as the gene responsible for the development of the Wilms kidney tumor that primarily affects children. The *WT1* gene is located at locus 11p13 and functions as a zinc finger-like transcription factor. Despite the initial classification of *WT1* as a suppressor gene, the overexpression of WT-1 in many cancers (breast cancer and acute leukemias) [79,80] has led to its recognition as an oncogene [81]. In a study conducted in 2004 [41], 48 out of 51 GBM samples (94%) showed positive staining for the WT-1 protein.**PTEN** (Phosphatase and tensin homolog): the *PTEN* gene is a suppressor gene located on 10q23. LOH (loss of heterozygosity) or methylation mutation disrupt the pathways that use phosphatidylinositol 3-kinase (PI3K) and are found in at least 60% of GBM cases [82]. Loss of PTEN function due to mutation or loss of heterozygosity (LOH) is associated with poor prognosis of GBM. PTEN is a protein with protein phosphatase and lipid phosphatase functions, and most of the onco-suppressive properties are due to the lipid phosphatase properties [83]. The PI3K/Akt pathway is blocked by PTEN, and loss of functional PTEN impairs the regulation of cell survival, cell growth, and proliferation [84]. According to Koul [82], loss of PTEN expression is indicative of the progression of a highly malignant tumor—PTEN is present in most low-grade tumors. Brito et al. [26] reported that PTEN deletion in *IDH*-wildtype GBM is associated with better OS.

## 4. Survival and Prognostic Factors

### 4.1. Incidence

The incidence of GBM shows minor locational variability (Figure 1). The data in Table 1 show the incidence of GBM by gender; unfortunately, due to different methodologies, they cannot be directly compared.

The location of the glioblastoma multiforme is concentrated in most cases in the frontal, temporal, and parietal lobes, and, less often, it affects other structures. In the last two decades, the increase in the number of detected cases (increase in morbidity/better diagnostic techniques) has been particularly noticeable, especially in the areas of the frontal and temporal lobes, [27]. In most reports, the incidence is similar to the work of Bohn et al., 2018 [28] and Tian et al., 2018 [85]—from the most common to the least common: frontal–temporal–parietal–occipital lobe–other structures of the brain. 

### 4.2. Age

Age is an important factor in the development of diseases such as cancer. Many studies confirm that age significantly affects the incidence of GBM, where the vast majority of cases occur in people over 40 years of age. This was confirmed by Kai et al. [32], where, for 47.9% of the subjects, the age of GBM diagnosis was ≥65 years, and 46.3% of the subjects were between 40 and 64 years. In the study by Tian et al. [85], far more cases of GBM (55.2%) were diagnosed among respondents between 41–60 years of age (Figure 2).

Based on the Central Brain Tumor Registry of the United States (CBTRUS) reports in 2013, 2017, and 2020, it was observed that the incidence of GBM increases with age, peaking at 75–84 years and decreasing after 85 years (Figure 3). Gittleman et al. [92] showed that the incidence rate of GBM increases with age.

### 4.3. Survival

GBM is the most aggressive diffuse glioma of the astrocyte lineage and remains an incurable tumor. Table 2 presents survival data for GBM. In most cases, the median survival was less than 15 months. In the study by Lam et al. [24], the median survival was 20, with 46.9% of subjects surviving for up to 2 years after diagnosis. In contrast, Fabbro-Peray et al. [6], Gittelman et al. [92], and Ostrom et al. [101] observed that patients most often lived for up to 1 year after diagnosis.

### 4.4. Urban/Rural Socioeconomic Status

In the CBTRUS Statistical Report [14], the incidence in urban counties was 2.1% (*p* = 0.0715) higher than that in rural counties, at 3.17 cases per 100,000 population, (95% CI = 3.14–3.2) and 3.1 per 100,000 population (95% CI = 3.04–3.17), respectively.

In a study conducted in the United States by Cote, Ostrom, et al. [110], glioma incidence was higher in counties with a higher socioeconomic status compared to counties of lower socioeconomic status. Counties of high socioeconomic status also had lower glioma mortality. The authors associated the differences in incidence and mortality with race and socioeconomic status, rather than the area of residence (urban vs. rural).

In 1976, Barker et al. [111] reported the incidence of glioma in the south of England at 3.94 per 100,000, which was lower in large urban areas. In contrast, E. V. Walker et al. [112] described lower mortality during the first 5 weeks of a GBM diagnosis in a group of people living in rural areas compared to those living in urban areas (hr: 0.86; 95% CI: 0.79 to 0.99). Higher mortality during the first 1.5 years after GBM diagnosis was also observed in the low-income group compared with the high-income group (hr: 1.15; 95% CI: 1.08 to 1.22).

## 5. Protective Factors

### 5.1. Gender and Hormones

The protective effects of female sex hormones on the development of GBM tumors, and their effect on increasing the incidence of meningiomas, are fairly well documented in the literature. Cowppli-Bony et al. [71], in a review on female sex hormones and the incidence of gliomas, observed an increased risk of developing these cancers in women with late first menstruation and late menopause, and a decreased risk in users of oral hormonal contraception and hormone replacement therapy (but the duration of use was not significant). In contrast, Michaud, Dominique S. et al. [113] and Wigertz et al. [114] reported no effect of the level of exogenous estrogen on the risk of GBM. Table 3 shows the incidence rate of GBM in both sexes. All studies presented indicate a higher incidence of GBM in men.

### 5.2. Non-Steroidal Anti-Inflammatory Drugs and Paracetamol

The proven protective effect of acetylsalicylic acid in inflammatory bowel disease on colorectal cancer risk [122] and the potential beneficial effect on other cancers encourages the evaluation of the effect of nonsteroidal anti-inflammatory drugs (NSAIDs) on the risk of developing GBM. The potential protective effect of NSAIDs in brain tumors is likely due to the inhibitory effect on prostaglandin E_2_ (PGE_2_) synthesis, a prostaglandin that may play a role in direct mutagenic effects, tumor growth, invasion, metastasis, immunosuppression, and angiogenesis [123,124]. Additionally, the potential role of elevated PGE_2_ levels associated with malignant brain tumors is evidenced by the decrease in the PGE_2_ levels after the surgical removal of a malignant brain tumor [125].

Altinoz, Meric et al. [126] described the effects of aspirin and its metabolites on GBM cells. Gentisic acid (GA), a metabolite of acetylsalicylic acid, blocks the attachment of fibroblastic growth factor (FGF) to its receptor, and the sulfonate metabolite dobesilic acid blocks the growth of C6 GBM cell line cells in vivo.

However, the molecular rationale behind the protective effect of NSAIDs on GBM development is not unequivocally reflected in the literature. Publications vary in their results and are often inconclusive due to many potential confounding variables and due to a failure to obtain adequate numbers of GBM patients. Collecting reliable data on NSAID use is a major challenge, especially because of the cognitive impairment during GBM treatment and the natural course of this neuroproliferative disease. In addition, the high heterogeneity of GBM documented in the Cancer Genome Atlas (TCGA) [127,128] and its intricacy may influence the different responses of different tumor subtypes to NSAIDs and result in discrepancies.

Scheurer et al. [129] described a 20% reduction in GBM risk after using NSAIDs, but the confidence interval was very close to 1.00. The studied subjects with developed glioma were less likely to report regular aspirin use in the past (OR: 0.69; CI: 0.56, 0.87). A case–control study published in 2004 by Sivak-Sears et al. [130] compared San Francisco Bay Area residents with GBM (*n* = 236) to a matched control group (*n* = 401). Based on the interviews collected, the GBM group reported consuming at least 600 fewer NSAID tablets in the 10 years prior to their disease than the control group (OR = 0.53, 95% CI: 0.3–0.8). Another case–control study conducted between 2007 and 2010 with a group of 517 GBM patients and 400 participants in the control group showed an inverse relationship between NSAID use for at least 6 months and the risk of developing GBM (OR = 0.68, 95% CI 0.49–0.96) [131].

In the literature, some publications argue a lack of correlation between NSAID use and the risk of glioma or GBM. A prospective study initiated in 1995 on a group of 302,767 US residents by Daugherty et al. [132] described the occurrence of 341 cases of gliomas in that group, including 264 GBM cases. Regular aspirin use (more than twice a week) was not associated with the risk of GBM (HR = 1.17; 95% CI, 0.83–1.64) compared with non-use. For the use of the rest of the NSAIDs, excluding aspirin, there was also no correlation with the occurrence of GBM (HR = 0.83; 95% CI, 0.56–1.20) compared with no NSAID use.

A Danish case–control study of a large group of patients (*n* = 2688) and a control group of *n* = 18,848 based on the Danish Cancer Registry and NSAID prescriptions from 2000 to 2009 also found no correlation between NSAID use and glioma risk. Bruhns et al. [133] showed no statistically significant difference in OS between patients using and not using NSAIDs in therapy (*p* = 0.75; 95% CI: 10.12–18.13).

### 5.3. Other Medications

#### 5.3.1. Antihistamines

Scheurer et al. [129] observed an OR of 0.89 (95% CI 0.63, 1.25) for GBM in antihistamine users. In a subsequent study, Scheurer et al. [134] observed that antihistamine use was associated with a 37% increased risk of glioma (odds ratio 1.37; 95% CI: 0.87; 2.14). In contrast, a 3.56 times increase in risk was observed in patients reporting a history of asthma or allergies who used antihistamines regularly for more than 10 years.

Schlehofer et al. [135] found a 30% reduction in the risk of glioma in adults using antihistamines. Schoemaker et al. [136] also reported a slight, but non-significant, reduction in risk associated with the use of antihistamines by those reporting conjunctivitis, allergic rhinitis, and hay fever.

McCarthy et al. [137] noticed that oral antihistamine use was inversely associated with glioma risk, although the adjusted OR was not statistically significant for those with low-grade glioma.

#### 5.3.2. Statins

A meta-analysis by Xie et al. [138] showed no lengthening of OS and PFS in statin users. In the study by Ferris et al. [131], taking statins showed a statistically significant inverse relationship between the duration of therapy and glioma risk.

Cote et al. [139] observed that associations between statin use and the risk of glioma were similar in the combined cohorts (HR = 1.30, 95% CI 0.99–1.69), and were statistically significant among men (HR = 1.58, 95% CI 1.06–2.34), but not among women (HR = 1.10, 95% CI 0.77–1.58).

A meta-analysis by Rendom et al. [140] showed that statins were potent anti-cancer drugs that suppressed glioma growth through various mechanisms in vitro. However, these effects were not statistically significant in terms of glioma incidence and survival.

#### 5.3.3. Cannabinoids

Cannabinoids originally referred to the bioactive components of the Cannabis sativa plant, namely the psychoactive cannabinoid Δ^9^-tetrahydrocannabinol (THC) and other phytocannabinoids, e.g., cannabinol, cannabidiol (CBD), and cannabigerol, or the flavor and aroma agent β-caryophyllene (BCP) [133]. Most cannabinoids bind to G protein-coupled cannabinoid receptors, CB_1_ and CB_2_, which act as agonists or inverse agonists. Importantly, GBM tumors express both major cannabinoid-specific receptors (CB_1_ and CB_2_), and the expression of these receptors has been detected in GBM cell lines, in ex vivo primary tumor cells from GBM patients, and in situ in GBM tissue biopsies. It has been observed that highly malignant gliomas, such as GBM, express high levels of CB_2_, which is positively correlated with the degree of malignancy [141]. In view of this fact, it was considered that cannabinoid receptor agonists could be used as anticancer agents [142]. This was confirmed by Guzman et al. [143], who reported a reduction in cancer cell proliferation after THC administration in two of nine patients. In contrast, several in vivo studies demonstrated that cannabinoids can significantly reduce tumor volume in orthotopic and subcutaneous animal models of glioma [144].

Three major mechanisms have been observed to inhibit tumor growth:Apoptosis and cytotoxic autophagy;Mechanisms that inhibit cell proliferation;Anti-angiogenic mechanisms.

This was confirmed by a 2013 review by Ellert-Miklaszewska [141], in which the authors described the occurrence of apoptosis involving proteins from the Bcl-2 family in response to treatment with cannabinoids. Hernandez-Tiedra et al. [145] observed that cannabinoids alter the permeability of the membranes of autophagosomes and autolysosomes, causing the release of cathepsins in the cytoplasm, which activates cell death by apoptosis. In contrast, Massi et al. [146] reported that cannabinoids induce apoptosis through oxidative stress. In addition to cancer cell apoptosis, cannabinoids can induce cell cycle arrest and thus inhibit cancer cell proliferation [147].

#### 5.3.4. Atopy

Lehrer et al. [148] showed, in their study taking into account 1p19q deletion, TERT overexpression, and *TP53* and *ATRX* mutations, that OS was greater in subjects with allergies (*p* = 0.025, HR 0.525, 95% CI 0.299–0.924). Schwartzbaum et al. also described the protective effect of asthma-associated gene polymorphisms on the risk of GBM [149].

There is an inverse relationship between atopy and the risk of developing glioma cancers.

A meta-analysis conducted in 2007 by Linos et al. [150] on 3450 patients with gliomas showed that the pooled relative risks (RRs) of glioma incidence in people with atopy compared to those without a history of atopy for allergy, asthma, and eczema were 0.61 (95% CI = 0.55 to 0.67), 0.68 (95% CI = 0.58 to 0.80), and 0.69 (95% CI = 0.58 to 0.82), respectively.

The mechanism responsible for these relationships has not been clearly established. A plausible explanation for this phenomenon is an as-yet unexplained increased sensitivity of the immune system. Allergy has been implicated in tumorigenesis, not only in gliomas, but also in pancreatic cancer. Interestingly, some IgE antibodies directed against allergens cross-react with glioma antigens [151], and some authors have reported an inverse correlation between the plasma antibody levels and glioma risk [150,151].

## 6. Risk Factors

### 6.1. Tobacco Smoking and Nitrosamines

Cigarette smoking has not been clearly linked to an increased risk of developing GBM [152,153] and glioma [152,153,154]. Because of the mixed findings, further attempts to establish a correlation or lack thereof are desirable, especially because cigarette smoke is a proven risk factor for the development of malignancies in certain organs. Cigarette smoke mutagens, such as tobacco-specific nitrosamines (TSNAs) and polycyclic aromatic hydrocarbons (PAHs), penetrate the blood–brain barrier [155], which may potentially affect the development of central nervous system tumors [156]. Considerable scientific evidence also points to the carcinogenic effect of TSNA in causing malignancies of the lung, pancreas, esophagus, and oral cavity. The most recent International Agency for Research on Cancer (IARC) monograph did not classify the nervous system as an organ in which carcinogenesis is caused by tobacco products [156].

Nitrosamines can originate from cigarette smoke, but also from the reaction of nitrates and nitrites used in meat products—hams, bacon, and sausages. N-nitrosodimethylamine (NDMA) is one of the most common nitrosamines in food [157,158,159]. NDMA is a potent carcinogen capable of inducing cancer in animal models [160]. Nitrates present in food entering the digestive system are absorbed into the blood and then secreted into the saliva. Following ingestion, they are passed into the stomach, where they are converted to nitrosamines in an acidic environment [161]. A study of patients diagnosed between 1987 and 1991 in Israel found that N-nitroso compounds were not directly linked to brain tumors [162].

In a study by Michaud et al. [163], neither the group consuming the most processed meat products nor the nitrate-exposed group had an increased risk of glioma (RR: 0.92; 95% CI: 0.48, 1.77 and RR: 1.02; 95% CI: 0.66, 1.58, respectively).

A meta-analysis by Saneei et al. [164] included data from 18 observational studies and found no association between the consumption of processed red meat and increased incidence of glioma.

### 6.2. Race/Ethnicity

There is a limited association between specific ethnic groups and the risk of developing GBM. Ostrom et al. [100] reported a 2.97 times higher incidence of GBM in Caucasians compared to Asians, and a 1.99 times higher incidence in Caucasians compared to African Americans.

A 2006 study by Fukushima et al. [165] compared mutations found in primary GBM in a Japanese group with mutations found in the Swiss group described by Ohgaki et al. [99]. The results of the study by Fukushima et al. [165] suggest high molecular similarity of GBM, despite the different genetic backgrounds of Asians and Caucasians (Table 4).

### 6.3. Ionizing Radiation

Ionizing radiation is a recognized risk factor for many cancers. Direct damage to genetic material or the generation of free radicals in the vicinity of DNA strands results in an increased incidence of mutations within the genetic material of cells. Since controlled clinical trials on the effects of radiation on carcinogenesis are not feasible for ethical reasons, case–control studies play a major role in describing this phenomenon. Ron et al. [167] already in 1988 linked doses of 1–2 Gy to an increased risk of neuronal tumors. A literature review by Bowers et al. [168] in 2013 documented an 8.1–52.3 times increased risk of central nervous system cancer after radiotherapy to the head for a CNS tumor in childhood compared to the general population, proportional to dose.

Most studies on the relationship between computed tomography (CT) and the risk of glioma development in children have not shown an increased risk, apart from a study describing one excess brain tumor per 10,000 patients over a 10-year period after exposure to one CT scan [169].

### 6.4. Head Injury

Because of the described anecdotal cases of CNS tumors (not just GBM) being diagnosed after head trauma, further studies on head trauma as an etiologic factor of brain tumors have been conducted, with mixed results. Unfortunately, the available research is quite limited. Proving a causal relationship is very difficult in this case [170]. In a study on the Danish population, gliomas were not diagnosed more frequently in patients after head injury—the standardized incidence ratio (SIR) after the first year was 1.0 for glioma (CI = 0.8–1.2) compared to the general Danish population. Tumors detected during the first-year period were not considered due to the detection of incidental lesions already existing during the trauma [171]. A study conducted in 1980 showed an increased odds ratio (odds ratio = 2.0, *p* = 0.01) in women compared with the control group in the incidence of meningiomas following head trauma [172]. In contrast, a case–control study evaluating the incidence of meningiomas and gliomas after head injury documented a higher risk of meningiomas, but a lower risk of gliomas (OR = 1.2, 95% CI: 0.9–1.5 for any injury; OR = 1.1, 95% CI: 0.7–1.6) [173]. Potential problems with the study may include the use of diagnostic methods using ionizing radiation, which is a proven risk factor for cancer, and potential problems with recalling past injuries and the non-standardized assessment of their extent.

### 6.5. Obesity

Adipose tissue has many functions in the human body. In addition to storing nutrients in the form of fats, it has a secretory role, for example, secreting estrogens [174] and pro-inflammatory substances [175,176]. For these reasons, it may have a potential impact on the development of cancer, including GBM.

Current data suggest that low body weight (BMI < 18.5 kg/m^2^) at age 21 is associated with a lower risk of developing gliomas later in life, although the results were only statistically significant in the group of women [177]. Moore et al. [178] found that patients who were obese at age 18 (BMI 30.0–34.9 kg/m^2^) had nearly four times the risk of developing gliomas compared to those who had a BMI of 18.5–24.9 kg/m^2^ at age 18 (RR = 3.74; 95% CI = 2.03–6.90; *p* trend = 0.003).

In the study by Kaplan et al. [162], increased fat and cholesterol consumption was inversely related to the incidence of glioma (high fat intake OR = 0.45, 95% Cl 0.20–1.07; high cholesterol intake: OR = 0.38, 95% Cl 0.14–1.01). Cote et al. [174] observed an inverse relationship between hyperlipidemia and glioma.

A study on a group of patients diagnosed between 1987 and 1991 in Israel found a relationship between the occurrence of gliomas and meningiomas and a protein-rich diet (OR = 1.94, 95% CI 1.03–3.63) [162]. Wiedmann et al. [119] did not observe an increased risk of glioma in overweight or obese individuals.

Seliger et al. [179] described a decrease in the risk of GBM in people with diabetes (OR = 0.69; 95% CI = 0.51–0.94). The decrease in risk was most pronounced in men with more than 5 years of disease or with poor glycemic control (HbA1c ≥ 8). In contrast, the effect of lower GBM risk was absent in women (OR = 0.85; 95% CI = 0.53–1.36).

### 6.6. Growth

Although a tall stature is associated with a higher incidence of certain cancers [180,181], the exact mechanism of this phenomenon has not been explained. It is likely that the insulin-like growth factor (IGF) and growth hormone (GH) pathways, which determine growth and final height in humans, are involved. The IGF concentrations peak at puberty and then decline in the third decade of life [182]. More than 80% of GBM tumors overexpress insulin-like growth factor binding protein-2 (IGFBP-2), one of the biomarkers of GBM malignancy [183,184]. In less aggressive tumors, IGFBP-2 is usually undetectable and appears with tumor progression [185].

In the paper published by Moore et al. [178], the risk of developing glioma among tall people (over 190 cm) was twice as high as that among people less than 160 cm tall (multivariate relative risk [RR] = 2.12; 95% confidence interval [CI] = 1.18–3.81; *p* trend = 0.006). In contrast, a study by Little et al. [176] did not link adult height to the risk of developing glioma.

### 6.7. Metals

The International Agency for Research on Cancer (IARC) lists cadmium, cadmium compounds, chromium compounds, and nickel compounds as human carcinogens, with lead as a potential carcinogen. None of these have been found to be associated with brain tumors. The ability of some heavy metals to penetrate the blood–brain barrier and to enter through the olfactory nerve pathway [186] prompts a closer examination of their effects on the risk of GBM.

A study conducted in 1970 examining job-exposure matrix (JEM)-based exposures to individual metals did not observe an increased risk of glioma in relation to occupational exposure to chromium, nickel, or lead among 2.8 million male workers (*n* = 3363 cases of glioma).

Parent et al. [187] reported an increased incidence of glioma associated with occupational exposure to arsenic, mercury, and petroleum products. However, they did not report an increased OR for glioma for welders exposed to lead, cadmium, or welding fumes [187]. Lead may also induce oxidative stress and disturbances in energy metabolism, induce apoptosis, and affect certain signaling pathways [187,188,189,190,191]. A meta-analysis by Ahn et al. [192] reported an increased risk of malignant brain tumors associated with lead exposure (pooled OR = 1.13, 95% CI: 1.04–1.24). Rajaraman et al. [193] observed no relationship between lead exposure and glioma risk.

Bhatti et al. [191] examined the potential carcinogenicity of lead by analyzing the modification of single-nucleotide polymorphisms (SNPs) within genes functionally related to oxidative stress. The study included 494 controls, 176 GBM patients, and 134 meningioma patients who were evaluated for occupational lead exposure. *Rac family small GTPase 2* (*RAC2)* and *glutathione peroxidase 1* (*GPX1)* gene polymorphisms significantly modified the relationship between cumulative lead exposure and GBM risk.

### 6.8. Nutritional Factors, Chemicals, and Pesticides

Brain tissue necrosis associated with GBM invasion leads to the release of triglycerides and may be accompanied by the release of toxins co-stored in phospholipid-rich neural tissue [194].

In a 1992 study using data from the Canadian National Cancer Incidence Database and Provincial Cancer Registries, Morrison et al. [195] found a statistically significant relationship between the risk of death from GBM and increased exposure to fuel/oil emissions (test for trend *p* = 0.03, RR for highest-exposure quartile was 2.11, 95% confidence interval = 0.89–5.01). They further suggested inverse associations of cholesterol and fat consumption with brain tumor risk, which they described as inconsistent with other studies [162].

In a study on T98G and U138-MG GBM cells, researchers attempted to determine the cytotoxic or proliferative effects of chemical compounds. The proliferative effect occurred only for the T98G line with perfluorodecanoic acid (PFDA), perfluoroacetate sulfonate (PFOS), and testosterone. However, perfluorinated salt (ammonium perfluoroacetate) and dehydroepiandrosterone (DHEA) showed no proliferation-stimulating effect, suggesting that the proliferative effect is not mediated by androgen receptor activation. The authors concluded that exposure to certain substances released during necrolysis may affect the subsequent growth of GBM and the adoption of more aggressive forms of GBM [194].

An in vitro study subjected the U87 GBM cell line to long-term exposure to low doses of a mixture of pesticides (chlorpyrifos-ethyl, deltamethrin, metiram, and glyphosate). Exposure resulted in the development of resistance to chemotherapeutics (cisplatin, temozolomide, 5-fluorouracil, among others) and increased expression of ATP-binding cassette (ABC) proteins [196].

Kuan et al. [197] reported weak or null associations between food groups, nutrients, or dietary patterns and glioma risk. They found no trends of decreasing glioma risk with increasing intake of total fruit, citrus fruit, and fiber, and a healthy diet.

### 6.9. Coffee and Tea

Coffee and tea may have potential cancer-protective effects. The presence of antioxidants, such as polyphenols, caffeic acid, diterpenes (including kahweol and cafestol), and heterocyclic compounds [198,199,200,201], could explain the molecular basis for this finding. A study by Kang et al. [201] reported the inhibition of GBM cell growth in vitro after exposure to caffeine by the inhibition of inositol trisphosphate receptor subtype 3. Polyphenol (2)-epigallocatechin-3-gallate restores the expression of methylated (silenced) genes in cancer cells, including MGMT, a protein with a DNA repair function [202]. Huber et al. [203] described elevated MGMT protein levels in rat livers after exposure to Kahweol and Cafestol (diterpenes).

Studies on the effects of coffee and tea on glioma risk are inconclusive. Holick et al. [204] reported an inverse relationship between caffeine consumption and glioma risk among men, but not among women. In contrast, in a cohort of 545,771 participants, Dubrow et al. [205] found no reduction in glioma risk with increased coffee and tea consumption. However, in a full multivariate model, there was an almost statistically significant inverse relationship between the highest level of tea consumption (three cups per day) and glioma risk (HR = 0.75; 95% CI, 0.57–1.00).

In a more recent study on a British population cohort (2,201,249 person-years and 364 GBM cases), Creed et al. [206] observed an inverse relationship between tea consumption and glioma risk that was statistically significant for all gliomas, and for GBM in men. In the same year, Cote et al. [207] published a paper using data from the Nurses’ Health Study (NHS), Nurses’ Health Study II (NHSII), and Health Professionals Follow-Up Study (HPFS) (6,022,741 person-years; 362 cases of GBM). The authors did not observe a relationship between coffee consumption and glioma risk, but noted a borderline inverse relationship between tea consumption and glioma risk for the combined cohort of men and women (HR for >2 cups/day versus <1 cup/week 0.73, 95% CI: 0.49–1.10, *p*-trend = 0.05).

Michaud et al. [208] observed a statistically significant inverse relationship between coffee intake and glioma risk in a group consuming 100 mL or more of coffee or tea per day compared to a group consuming less than 100 mL of coffee or tea per day. Based on the six studies included in the meta-analysis, Malerba et al. [209] suggested no association between coffee or tea consumption and the risk of glioma, but their work had limitations due to the small number of papers analyzed.

### 6.10. Alcohol Use

Alcohol can cross the blood–brain barrier and, therefore, can affect glial cells; in addition, it is a recognized risk factor in multiple cancers [210]. The metabolism of alcohol (at higher concentrations in the body) produces acetaldehyde and reactive oxygen species that have toxic effects on cells; acetaldehyde has been shown to be neurocarcinogenic in animals [211]. Additionally, alcoholic beverages contain N-nitroso compounds that cause brain tumors in animals [211,212,213]. Despite this, the study by Qi et al. [214] based on 19 meta-analyses reported no association between glioma incidence and alcohol consumption. These observations were confirmed by a recent study by Cote et al. [215], who even indicated that low to moderate alcohol consumption may reduce the risk of glioma.

### 6.11. Sleep and Melatonin

Samatic et al. [216] noticed that sleep duration is not linked with the risk of glioma. Oreskovic et al. [217] reported that there are mechanisms of pro-tumor effects of sleep disorders, including phase shifts, decreased antioxidant levels, immunosuppression, metabolic changes, melatonin deficiency, cognitive impairment, or epigenetic changes. All of these changes significantly affect the poorer prognosis of patients with malignant brain tumors and are potential exacerbating factors for tumor progression. In addition, the occurrence of a brain tumor contributes to sleep disorders.

Lissoni et al. [218] evaluated the effects of melatonin co-treatment in patients with GBM undergoing radical or adjuvant radiotherapy. They observed that the patient survival percentage of the RT and melatonin group was significantly higher than that of the RT alone group (6/14 vs. 1/16 patients).

Cutando et al. [219] reported that melatonin administration reduces the incidence of malignant tumors in vivo and increases the survival time of patients with GBM treated by radiotherapy. A study by Martin et al. [220] showed that melatonin sensitizes human malignant glioma cells against TRAIL-induced cell death. Furthermore, the melatonin/TRAIL combination significantly increases apoptotic cell death compared to TRAIL alone. A study by Zheng et al. [221] confirmed the anti-glioma function of melatonin to be mediated partly by suppressing glioma stem cell (GSC) properties through EZH2-NOTCH-1 signaling.

### 6.12. Inflammation

Even in a healthy body, gene mutations can lead to tumorigenesis and GBM. Numerous mechanisms are in place to offset these processes so that altered cells are effectively destroyed by the immune system before tumor formation [222,223]. Even when a tumor forms, the immune system can destroy it at an early stage. A molecular mechanism that facilitates this process is inflammation [224]. Chronic inflammation, on the other hand, can facilitate tumor formation [225] by damaging DNA, resulting in mutations and tumorigenesis [225,226]. Additionally, chronic inflammation triggers mechanisms that can inhibit an otherwise robust immune system response [227] and thus inhibit the immune system from fighting newly formed cancer cells. For this reason, the factors that trigger this physiological state will increase susceptibility to cancer. Some of the best-studied inflammation-related factors involved in GMB are tumor necrosis factor α (TNFα) and interleukins 1 and 6 (IL-1 and IL-6).

Tumor necrosis factor α (TNFα) is a soluble cytokine involved in directing the systemic inflammatory response [228]. It can exert antitumor effects on glioma cells, but can also enhance tumor progression. TNFα can facilitate angiogenesis by increasing epidermal growth factor receptor (EGFR) activity [229]; it induces immune cell suppression through the activation of the NF-κB and STAT3 pathways [230], and decreases the expression of the tumor-suppressor gene PTEN in glioma [231]. Our results suggest that TNFα is involved in reduced macrophage infiltration, suggesting that TNFα plays a suppressive role by demonstrating the ability to promote tumorigenesis [232]. Since abnormal epidermal growth factor receptor EGFR signaling is widespread in GBM, EGFR inhibition seemed to be a promising therapeutic strategy. However, EGFR inhibition in GBM causes a rapid upregulation of TNFα, which in turn leads to the activation of the JNK-Axl-ERK signaling pathway involved in resistance to EGFR inhibition [233]. A previous study showed that TNFα induces the upregulation of angiogenic factors in malignant glioma cells, which plays a role in RNA stabilization [234]. This confirms that TNFα in GBM cells may play an important role in tumor progression.

Interleukin 1 (IL-1) is a potent inducer of proangiogenesis and proinvasion factors, such as VEGF, in human astrocytes and glioma cells. IGF2 induction [235,236] is strongly stimulated by IL-1 in astrocytes [237]. IL-1 is also a major inducer of astrocyte/glioma miR-155, a microRNA involved in inflammation-induced cancer formation [238]. The specific microRNA (miR-155) targets cytokine signaling suppressors, potentially leading to the overactivation of STAT3, a transcription factor important in glioma progression.

IL-1α has been implicated in cancer pathogenesis, but there is little evidence of its role in GBM. To date, its function has been shown to be both pro- and anti-tumor in various cancer types [239]. IL-1α secretion by tumor cells causes the constitutive activation of NF-κB, which results in the expression of genes involved in the cascade of metastatic processes and angiogenesis [240].

GBMs have been shown to produce large amounts of IL-1β, which plays a key role in glioma aggressiveness and survival. IL-1β is a major pro-inflammatory cytokine that triggers a number of tumorigenic processes by activating various cells to upregulate key molecules involved in oncogenic events. Elevated levels of IL-1β have been observed in cultures of GBM cell lines [241] and in samples from human GBM tumors [242]. IL-1β receptor (IL-1R) is found in GBM cells and tissues [243]. The binding of IL-1β to IL-1R activates a cascade of NF-κB and mitogen-activated protein kinase (MAPK) signaling pathways [244]. IL-1β-induced ERK activation can also have mitogenic effects on human glioma U373MG cells and significantly increase GBM cell proliferation [245]. IL-1β-dependent activation of the NF-κB, p38 MAPK, and JNK pathways in GBM cells also leads to increased expression of VEGF, which promotes angiogenesis, migration, and invasion [246]. In addition, IL-1β-mediated up-regulation of factor HIF-1 [247] is involved in molecular responses to hypoxia, which is a key component of GBM progression.

The glioma environment is subject to chronic inflammation, and IL-6 is one of the cytokines strongly associated with the chronic inflammatory phenotype often associated with GBM. Tumor-associated macrophages make up a large majority of noncancerous cells in tumors and are major producers of IL-6 [248]. Interleukin 6 (IL-6) has been shown to be a factor involved in the malignant progression of glioma [249]—it promotes regeneration, invasion, and angiogenesis. In glioma, the elevated expression of IL-6 and its receptor is associated with poor patient survival [250]. IL-6 promotes tumor survival by suppressing immune surveillance through the recruitment and stimulation of tumor-associated myeloid-derived suppressor cells and neutrophils. This paralyzes the response of surrounding type-1 helper T cells and cytolytic T cells, ultimately leading to T cell dysfunction and the inhibition of tumor cell clearance. IL-6 is specifically involved in GBM as the stimulation of brain tumor cells by IL-6 promotes three major signal transduction pathways involved in gliogenesis—(1) p42/p44-MAPK, dysregulated in approximately one-third of all cancers and strongly involved in the detection and processing of stress signals [251]; (2) PI3K/AKT, a signaling pathway associated with enhancing angiogenesis, activating the EMT transition to increase invasion, and promoting metastasis [252]; and (3) JAK-STAT3, a pathway that blocks tumor recognition by immune cells and promotes cell cycle progression and the inhibition of apoptosis [253].

### 6.13. Electromagnetic Radiation

With the popularization of electronic devices, such as microwave ovens and cell phones, the impact of exposure to electromagnetic waves and the risk of developing CNS tumors became a controversial topic. The impact of phones on tumor development remains inconclusive due to the mixed results from studies, the relatively short time since the prevalence of smartphones, and the numerous confounding factors in the research.

Today, people are commonly exposed to radio-frequency electromagnetic fields (RF-EMF) (30 kHz–30 GHz) through electronic devices, such as cell phones, cordless phones, radios, and Bluetooth. These devices are located in close proximity to users so that even low-power transmitters are not precluded from potential effects on health. The specific RF energy absorption rate (SAR) of the most common source, mobile telephones, is influenced by many factors, such as the design of the device, the position of the antenna in relation to the user’s head, the anatomy of the user’s head, how the phone is held, and the quality of the connection between the cell phone and the network station. A working group [254] in 2011 concluded that, despite the high risk of error in the available studies, the potential carcinogenic effects of RF-EMF cannot be ruled out.

A pooled analysis of Swedish case–control studies of people who had used cell phones for more than 25 years was conducted by Hardell and Carlberg [255], showing that the OR of developing glioma was 3.0 (95% CI: 1.7–5.2). In contrast, Villeneuve et al. [256] suggested that the lack of increase in glioma incidence rates with the increasing popularization of cell phones supports the lack of a causal relationship.

In a study published in 2010 [257], a group who used a cell phone at least once a week over a six-month period had a lower risk of developing glioma than the group who never used a cell phone (OR = 0.81 (95% CI: 0.70–0.94)), but the most exposed (10th decile (≥1640 h)) in terms of cumulative exposure had a 40% higher risk of developing glioma (OR = 1.40, 95% CI = 1.03–1.89). This indicates the possible presence of confounding factors, study biases, and suboptimal selection of study participants.

In studies on the effect of cellphone use on the survival of GBM patients, Olsson et al. [258] did not report any reduced OS compared to those who did not use cell phones regularly.

## 7. Treatment of GBM

Current treatment methods are based on a combination of surgical approaches along with radiotherapy and chemotherapy. Recent significant changes in therapeutic methods involve the addition of temozolomide (alkylating chemotherapeutics) to the treatment regimen. Currently, the determination of mutations of specific tumor cell genes has a high predictive and prognostic value and allows the development of targeted therapies.

GBM remains an incurable cancer [2]. The goal of medical management is to make a diagnosis by the biopsy of the tumor and to prolong and improve the quality of the patient’s life. A definitive diagnosis is made on the basis of a mandatory histopathological examination and molecular studies [259]. Most patients are treated with multiple-modality therapy. Symptomatic treatment (symptoms occurring as a result of local pressure damage to the centers—epilepsy, neurological losses, hydrocephalus, and elevated intracranial pressure (ICP)) is an important part of treatment. Surgical treatment remains the primary focus of care and patient management [260]. Tumor shedding makes it difficult to distinguish the tumor cell mass from the surrounding healthy tissue. For this reason, despite achieving complete macroscopic resectability, local recurrence does occur. Roh et al. [261] compared gross total resection (GTR) with lobectomy including the tumor and surrounding noncancerous tissue (SupTR), but did not report statistically significant differences in the postoperative Karnofsky Scale scores in the operated patients. The median PFS was longer in the SupTR group (30.7 months (95% CI 4.3–57.1)) compared to that in the GTR group (11.5 months (95% CI 8.8–14.2)), which was significant (*p* = 0.007). However, it should be mentioned that SupTR was only performed when the tumor was located in the frontal or temporal lobe and without damage to the motor cortex. The SupTR surgical intervention probably did not damage a significant portion of healthy neurons, but removed cells whose axons would have been damaged by GTR surgery anyway.

The failure to achieve clear surgical margins is due to the presumed progression of tumor cells along neuronal fibers (neuropils) without macroscopic changes, which does not give the operator a chance to achieve microscopic resectability and cure the patient with surgery alone. Extensive resection (or SupTR) has the limitation of potentially damaging nerve centers and pathways, where dysfunction dramatically decreases the patient’s quality of life and well-being. Currently, fractionated adjuvant radiotherapy is the standard of care after surgical resection [259].

Another form of GBM treatment is chemotherapy. In a paper published in 2005 in *NEJM*, Stupp et al. [261] reported greater efficacy of fractionated radiotherapy (fractionated focal irradiation in daily fractions of 2 Gy administered 5 days per week for 6 weeks, for a total of 60 Gy) combined with temozolomide administration (75 mg per square meter of body surface area per day, 7 days per week from the first to the last day of radiotherapy), followed by six cycles of adjuvant temozolomide (150 to 200 mg per square meter for 5 days during each 28-day cycle) compared with radiotherapy alone.

Furthermore, in this study, the efficacy of temozolomide was greater in the presence of a methylated *MGMT* promoter in GBM cells. Subsequent adjuvant temozolomide chemotherapy combined with radiotherapy (TMZ/RT → TMZ) significantly improved median survival, and 2- and 5-year survival and is currently the standard of care for patients with GBM under 70 years of age, or 70+ in good physical condition. TMZ is administered daily for the duration of the radiotherapy and then for 5 days every 4 weeks for six cycles as maintenance (adjuvant) therapy after the completion of radiotherapy. Mutation of the MGMT gene promoter has been shown to be the strongest predictive marker, and the greatest benefit of chemotherapy appears to occur in patients with this mutation.

Li X. et al. [262] linked GBM treatment to a higher incidence of subsequent cancers. This is probably due to the use of radiotherapy, with consequential mutagenic effects and the individual’s predisposition to develop tumors. The consequences of anticancer treatment also relate to the side effects of temozolomide. In rare cases, TMZ may result in the development of myelodysplastic syndrome, acute myeloid leukemia, and even acute lymphoblastic leukemia [263].

Currently, there is no screening tool or test to detect GBM before the onset of clinical symptoms. The gold standard for imaging studies showing GBM is MRI [264].

## 8. Conclusions

This review presents the pathogenesis of GBM, including the key molecular changes that trigger GBM. In addition, the risk factors for GBM, as well as factors with protective effects, were analyzed. Unfortunately, due to significant differences in the methodology of epidemiological reports, it proved impossible to compare some data, e.g., compare GBM incidence in various regions of the world. It would, therefore, be valuable to establish an international cancer registry to reliably document key data on GBM.

Analysis of the data collected in this article confirms that GBM is in the higher grade of primary brain tumors and is far more common in men. Moreover, the risk of being diagnosed with glioma increases with age, while median survival remains low, despite medical advances. Despite the alarming growing trend in the incidence of GBM, it is still difficult to directly determine the causes of its occurrence, which is why further research into the etiology and treatment of GBM tumors should continue. In addition, it is difficult to clearly determine what impact the use of stimulants, certain drugs (e.g., NSAIDs), or the use of cell phones has on the development of glioma. It is also worth noting that today’s treatments cannot cure GBM patients, but only slightly extend their total life span.

## 9. Limitation

Although GBM is the most common and aggressive primary brain tumor, in the past several decades, we have seen little progress in finding better treatment options for patients diagnosed with this cancer. The complexity of the disease, the heterogeneity or highly invasive potential of GBM tumors, and the fact that, until recently, GBM research received less funding compared to other tumor types have contributed to little advancement in the treatment of this disease. Due to the high complexity of GBM, it is important to conduct studies with high reproducibility. Unfortunately, available epidemiological reports or studies on the risk of GBM are characterized by significant differences in results, which is probably due to differences in intra-population sex ratios, insufficient sample size relative to population size, and different inclusion and exclusion criteria. Because of these important differences, there is significant difficulty in comparing studies. Our review covers most of the areas of GBM epidemiology and indicates that the existing literature on GBM has its strengths and weaknesses. We have established significant challenges in the epidemiology of GBM, including short patient survival, cognitive changes in patients that hinder the collection of data, and low incidence that makes it difficult to carry out a prospective study. All these factors significantly affect the possibility of understanding the epidemiology of GBM. Because of the specificity of this disease, it seems important to extend research in GBM epidemiology, e.g., by analyzing a broader range of risk factors or biomarker-specific morbidity.

## Figures and Tables

**Figure 1 cancers-14-02412-f001:**
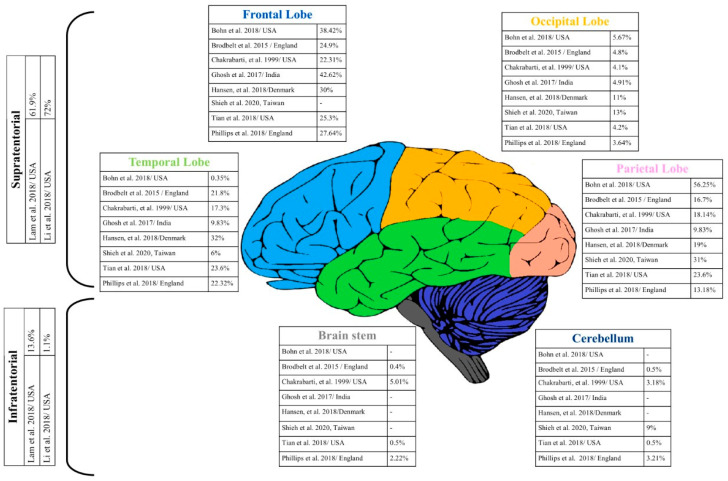
Summary of GBM locations in the central nervous system (%); some data also distinguished locations: Brain, NOS (not otherwise specified), Chamber NOS (not otherwise specified), Brain Overlapping Brain Damage, and Other, which are not shown in the figure; therefore, the sum of the percentages of the structures is not 100% [23,24,26,27,28,29,30,31,32]. own figure, no permission required for publication.

**Figure 2 cancers-14-02412-f002:**
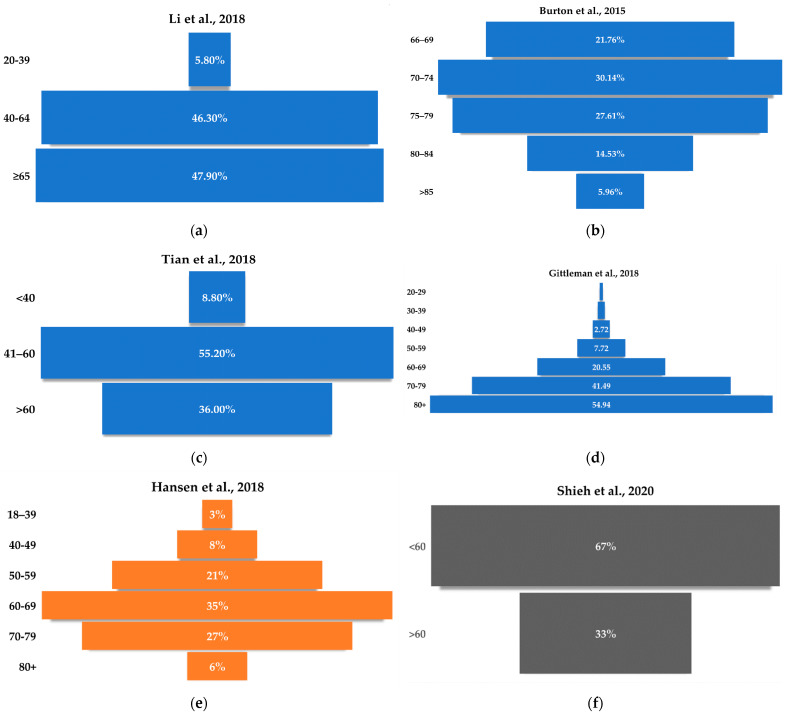
Age-related incidence of GBM in (**a**) [32], (**b**) [105], (**c**) [85] (**d**) [92] the USA, (**e**) Denmark [31], (**f**) Taiwan [106], (**g**) China [107], and (**h**) India [25].

**Figure 3 cancers-14-02412-f003:**
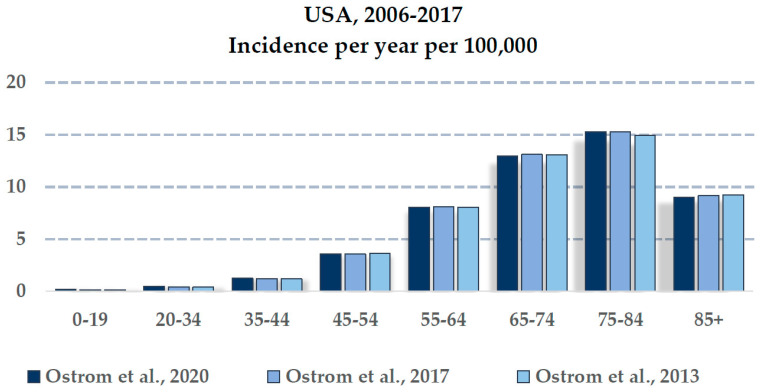
Age-related incidence of GBM (100,000/years) [14,100,101].

**Table 1 cancers-14-02412-t001:** Incidence rates of GBM for both genders.

Reference	Year Range	Country	Population Size (*n*)	Incidence/ 100,000/Year Male	Incidence/ 100,000/Year Female	Incidence/ 100,000/Year Male and Female	Age
Brodbelt et al., 2015 [29]	2007–2011	England	10,743	5.87 (5.73–6.02) ^b^	3.54 (3.44–3.65) ^b^	4.64 (4.56–4.73) ^b^	all
Chakrabarti et al., 2005 [30]	1974–1999	USA	3832	2.68 (95% CI 2.56–2.80) ^a^	1.67 (95% CI 1.59–1.75) ^a^	2.11 (95% CI 2.0–2.17) ^a^	≥20
Dho et al., 2017 [86]	2013	Republic of Korea	19	0.12 ^j^	0.17 ^j^	0.87 ^j^	0–19
			629	0.99 ^j^	0.78 ^j^		all
Dobec-Meić et al., 2006 [87]	1996–2004	Croatia	63	5.1 ^g^	4.6 ^g^	4.8, (95% CI 3.7–6.2) ^g^	≥18
Dobes et al., 2011 [88]	2000–2008	Australia	2197	-	-	3.96 (3.37–4.52) ^a^	all/not specified
Fabbro-Peray et al., 2019 [6]	2008–2015	France	2053	-	-	3.3 ^f^	all
Fleury et al., 1997 [89]	1983–1990	France	764	3.09 ^b^	1.94 ^b^	2.38 ^b^	all
Fuentes-Raspall et al., 2011 [90]	1994–2005	Spain	134	-	-	1.59 ^d^	all
Fuentes-Raspall et al., 2017 [91]	1994–2013	Spain	463	5.05 (4.45; 5.72) ^e^	3.44 (2.97; 3.96) ^e^	4.17 (95% CI 3.80; 4.57) ^e^	all
Gittleman et al., 2018 [92]	2000–2014	USA	150,399			4.40 (4.38–4.42 95% CI) ^a^	>20
Gousias et al., 2009 [93]	2005–2007	Northwest Greece	36	4.12 ^k^	3.26 ^k^	3.69 ^k^	all
Hansen et al., 2018 [31]	2009–2014	Denmark	1364	6.3 ^h^	3.9 ^h^	-	19–89
Ho et al., 2014 [94]	1989–2010	Netherlands	9504	3.2 ^b^	1.9 ^b^	2.5 ^b^	≥18
Jazayeri et al., 2013 [95]	2000–2009	Iran	3101	-	-	0.76 (0.70–0.83) ^g^	all
Jung et al., 2013 [96]	2010	Republic of Korea	523	0.89 ^j^	0.95 ^j^	0.77 ^j^	all
Li et al., 2018 [32]	1973–2014	USA	28,835	-	-	4.1 ^l^	>20
Larjavaara et al., 2007 [97]	2000–2002	Finland	154	-	-	2 ^d^	20–69
Natukka et al., 2019 [98]	1990–2006	Finland	2284	-	-	3.8 (95% CI 3.7–4.0) ^e^	all
Natukka et al., 2019 [98]	2007–2016	Finland	1776	-	-	3.5 ^e^	all
Ohgaki et al., 2004 [99]	1980–1994	Switzerland	715	3.32 (CI, 2.69–4.09) ^d^	2.24 (CI, 1.56–3.22) ^d^	-	all
Ostrom et al., 2013 [100]	2006–2010	USA	50,872	3.97 ^a^	2.53 ^a^	3.19 ^a^	all
Ostrom et al., 2017 [101]	2010–2014	USA	56,421	3.99 (3.95–4.03 95% CI) ^a^	2.52 (2.49–2.56 95% CI) ^a^	3.20 (3.17–3.23 95% CI) ^a^	all
Ostrom et al., 2020 [14]	2013–2017	USA	60,056	4.03 (95% CI 3.98–4.07) ^a^	2.54 (95% CI 2.50–2.57) ^a^	3.23 (CI (95% 3.20–3.25) ^a^	all
Schoenberg et al., 1976 [102]	1935–1964	Connecticut	1167	2.07 ^i^	1.51 ^i^	-	all
Walker et al., 2019 [103]	2009–2013	Canada	5830	4.91 (95% CI 4.75–5.08) ^c^	3.24 (3.11; 3.37) ^c^	4.06 (3.95; 4.16) ^c^	all
Wanner et al., 2020 [104]	2009–2012	Georgia	72	0.59 (0.42; 0.82) ^a^	0.42 (0.29; 0.59) ^a^	-	all

^a^ Age-adjusted using the 2000 US standard population, ^b^ European age-adjusted incidence, ^c^ Adjusted to the 2011 Canadian census age distribution, ^d^ Age-standardized to the world’s standard population, ^e^ Adjusted to the 2013 European standard population, ^f^ Crude rate, ^g^ No information, ^h^ Age adjustment not specified, ^i^ All age adjustments used the direct method and the population of the United States in 1950 as the standard, ^j^ Adjusted to Segi’s world standard population, ^k^ Adjusted to the Greek population, ^l^ Annual age-standardized incidence rates (ASRs; per 100,000 population), *n*—number of subgroup members.

**Table 2 cancers-14-02412-t002:** Survival time of GBM patients.

Reference	Year Range	Country	Population Size (*n*)	Survival Me (Months)	Survival (%).			
					1 Year	2 Years	5 Years	10 Years
Brodbelt et al., 2015 [29]	2007–2011	England	10,743	6.1	28.4	11.5	3.4	-
Cheo et al., 2017 [107]	2002–2011	China	107	15.1	-	23.5	-	-
Fabbro-Peray et al., 2019 [6]	2008–2015	France	2053	11.2	47.1	20.1	4.5	-
Fuentes-Raspall et al., 2017 [91]	1994–2013	Spain	463	-	24.0	-	3.3	-
Ghosh et al., 2017 [25]	2012–2014	India	61	8	19.15	3.27		
Gittleman et al., 2018 [92]	2000–2014	USA	33,469	-	39.5	16.9	5.4	2.7
Hansen et al., 2018 [31]	2009–2014	Denmark	1364	11.2	-	-	-	-
Lam et al., 2018 [24]	2000–2010	USA	302	20	-	46.9	-	-
Narita and Shibui, 2015 [108]	2001–2004	Japan	1489	15	-	-	10.1	-
Ostrom et al., 2017 [101]	2000–2014	USA	33,951	-	39.7	17.2	5.5	-
Yan Yuan et al., 2016 [109]	1992–2008	Canada	14,120	-	26.5	9.5	4.0	-

Me—median, *n*—number of subgroup members.

**Table 3 cancers-14-02412-t003:** Incidence rates of GBM presented for both genders.

Reference	Year Range	Country	Population Size (*n*)	M:F Ratio
Bohn et al., 2018 [28]	2010–2014	USA	3473	1.40
Brodbelt et al., 2015 [29]	2007–2011	England	10,743	1.66
Bruhn et al., 2018 [115]	2001–2012	Sweden	143	1.6
Burton et al., 2015 [105]	1997–2009	USA	3759	1.15
Chakrabarti et al., 2005 [30]	1974–1999	USA	3832	1.6
Cheo et al., 2017 [107]	2002–2011	China	107	1.55
De Witt Hamer et al., 2019 [116]	2011–2014	Netherlands	2382	1.63
Dobec-Meić et al., 2006 [87]	1996–2004	Croatia Varazdin County	63	1.12
Dobes et al., 2011 [88]	2000–2008	Australia	2197	1.6
Fabbro-Peray et al., 2019 [6]	2008–2015	France	2053	1.5
Ghosh et al., 2017 [25]	2012–2014	India	61	2.59
Gousias et al., 2009 [93]	2005–2007	Northwest Greece	36	1.25
Hansen et al., 2018 [31]	2009–2014	Denmark	1364	1.6
Helseth and Mork, 1989 [117]	1955–1984	Norway	2813	1.48
Jung et al., 2013 [96]	2010	Republic of Korea	523	1.22
Li et al., 2018 [32]	1973–2014	USA	28,835	1.35
Nomura et al., 2011 [118]	1995–2004	Osaka, Japan	713	1.25
Ostrom et al., 2017 [101]	2010–2014	USA	56,421	1.36
Ostrom et al., 2013 [100]	2006–2010	USA	50,872	1.57
Shieh et al., 2020 [106]	2005–2016	Taiwan	48	1.18
Tian et al., 2018 [85]	2000–2008	USA	6586	1.60
Wiedmann et al., 2017 [119]	1963–1975	Norway	3102	1.33
J.-C. Xie et al., 2018 [120]	2004–2015	USA	30,767	1.38
Zampieri et al., 1994 [121]	1986–1988	Italy	72	1.25

*n*—number of subgroup members, M:F ratio—Male:Female ratio.

**Table 4 cancers-14-02412-t004:** Relationship between race/ethnicity and the incidence of GBM.

Reference	Year Range	Country	Population Size *(n*)	Age	Race/Ethnicity (%)				
					White	Black	Hispanic	Asian	Unknown/Other
Bohn et al., 2018 [28]	2010–2014	USA	3473	≥18	83.21	5.90	5.53	5.36	-
Burton et al., 2015 [105]	1997–2009	USA	3759	>65	93.11	3.14	-	-	3.75
Cheo et al., 2017 [107]	2002–2011	China	107	13–85	-	-	-	Chinese 76.6 Malay 9.3 Indian 7.5	6.5
Li et al., 2018 [32]	1973–2014	USA	28 835	>20	91.4	4.7	-	-	3.9
Ostrom et al., 2013 [100]	2006–2010	USA	50,872	all	91.78	5.62	6.77	2.22	0.38
Ostrom et al., 2017 [101]	2010–2014	USA	56,421	all	91.11	6.12	7.34	2.38	0.39
Ostrom et al., 2020 [14]	2013–2017	USA	60,056	all	91.47	6.21	7.87	1.87	0.45
Xie et al., 2018 [120]	2004–2015	USA	30,767	all	89.6	5.5	-	-	4.7
**Reference**	**Year Range**	**Country**	**Population Size (*n*)**	**Age**	**Race/Ethnicity (Incidence per 100,000)**				
					**White**	**Black**	**Hispanic**	**Asian**	**Unknown/Other**
Chakrabarti et al., 2005 [30]	1974–1999	USA	3832	>20	Latino 1.83 (1.65–2.01 95% CI) Non-Latino white 2.53 (2.44–2.62 95% CI)	1.45 (1.27–1.62 95% CI)	-	-	-
Ostrom et al., 2013 [100]	2006–2010	USA	50,872	all	3.45 (3.41–3.48 95% CI)	1.67 (1.60–1.73 95% CI)	2.45 (2.36–2.53 95% CI)	1.67 (1.57–1.78 95% CI)	1.48 (1.26–1.72 95% CI)
Ostrom et al., 2017 [101]	2010–2014	USA	56,421	all	3.46 (3.43–3.49 95% CI)	1.79 (1.73–1.85 95% CI)	2.42 (2.35- 2.5095% CI)	1.47 (1.26–1.69 95% CI)	1.61 (1.52–1.70 95% CI)
Ostrom et al., 2020 [14]	2013–2017	USA	60,056	all	3.51 (3.45–2.54 95% CI)	1.77 (1.71–1.83 95% CI)	2.46 (2.39–2.53 95% CI)	1.18 (1.11–1.25 95% CI)	1.49 (1.30–1.69 95% CI
Shabihkhani et al., 2017 [166]	2001–2011	USA	21,184	-	5.1 (95% CI 5.0–5.3)	-	3.4 (95% CI 3.3–3.5)	-	-

*n*—number of subgroup members.

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
