# Peer review of "Epidemiology of Glioblastoma Multiforme–Literature Review"

_cancers, 2022, doi:10.3390/cancers14102412_

Round 1

Reviewer 1 Report

In this manuscript, Grochans et al. review the epidemiology of GBM, with detailed reference to the existing literature. The paper is well organized and relatively well written, and is easy to follow. The overall findings and included information is not new, and several similar reviews have been published in recent years, but this manuscript nevertheless provides a good foundation and appraisal of the existing literature.

Major comments:

  • The authors are missing sections on areas of GBM epidemiology that have been recently investigated. Although some of these findings are negative, negative findings are still of interest. For example, there is no discussion of alcohol use, sleep, or inflammation and glioma risk. Certainly the level of investigation/evidence in these areas at least matches some of the sections included by the authors, though I realize that inclusion of all areas of GBM epidemiology may be difficult.
  • It may be helpful to add a section on strengths/limitations near the end of the manuscript. Although the authors reference the strengths and limitations of the existing GBM literature, explicit acknowledgement of challenges in GBM epidemiology (e.g., short survival, cognitive changes that make case-control enrollment difficult, low incidence making prospective studies difficult, etc.) may be helpful to provide context for the overall findings. This would also provide a helpful area to describe possible future directions for GBM epidemiology (e.g., a greater inclusion of biomarker specific analysis of risk factors and incidence).

Minor comments:

  • It is a bit strange to begin the review stating that you will be including papers that do not include only the disease in question. While I agree that GBM-specific literature is limited and it is reasonable to include the glioma literature, this should be mentioned later in the introduction or discussion, rather than right at the outset.
  • “Most malignant” (introduction, line 42), is a strange phrase. Also present in the Conclusion. Tumors are either malignant or not. Higher grade may be a better descriptor.
  • Figure 1 is excellent, Figures 2 and 3 should be improved. These are copied straight from Excel with very small font that makes interpretation difficult.
  • Because Table 3 is present, Table 1 probably does not need to include incidence by gender, and can include overall only.
  • There are other references available for antihistamines and glioma, including McCarthy et al. 2011 and an additional paper by Scheurer et al. (2008), among others.
  • There are other references available for statins and glioma, including Cote et al. (2019), Rendon et al. (2022), which show no significant association.
  • The section on cannabinoids is quite long considering nearly all of the evidence for these is laboratory rather than population based.
  • I would argue that evidence for racial/ethnic disparities and glioma risk is quite strong. This section is also incorrectly titled “Ethnicity” when it should be “Race/Ethnicity”. Same comment for the title of Table 4.
  • Section 6.8 is titled nutritional factors but does not discuss much of the rather rich literature on nutritional intake and glioma risk. For example, Kuan et al. (2019) provide an excellent study of nutrition and glioma showing no significant associations. Other studies have examined nutrients (e.g., flavonoids, vitamins) and glioma risk with some interesting findings.
  • It would be reasonable to cite the original study by Stupp et al. in NEJM (2005) when discussing the Stupp regimen for GBM.

Author Response

10th May 2022

Dear Reviewers,

We take the liberty to thank you for the insightful and careful evaluation of our article entitled “Epidemiology of Glioblastoma Multiforme - Literature Review ” by Szymon Grochans , Anna Maria Cybulska, Donata SimiÅ„ska, Jan Korbecki, Klaudyna Kojder, Dariusz Chlubek and Irena Baranowska-Bosiacka.

The comments helped us to improve the quality of the manuscript. We considered all comments and recommendations and responded to questions.

The correction throughout the manuscript were marked in yellow.

Our responses to the reviews are attached below.

Response to Reviewers

Comments and Suggestions for Authors

In this manuscript, Grochans et al. review the epidemiology of GBM, with detailed reference to the existing literature. The paper is well organized and relatively well written, and is easy to follow. The overall findings and included information is not new, and several similar reviews have been published in recent years, but this manuscript nevertheless provides a good foundation and appraisal of the existing literature.

MAJOR COMMENTS:

  1. The authors are missing sections on areas of GBM epidemiology that have been recently investigated. Although some of these findings are negative, negative findings are still of interest. For example, there is no discussion of alcohol use, sleep, or inflammation and glioma risk. Certainly the level of investigation/evidence in these areas at least matches some of the sections included by the authors, though I realize that inclusion of all areas of GBM epidemiology may be difficult.

RESPONSE:  Thank you for this suggestion.  We added discussion of alcohol use, sleep and inflammation and glioma risk. 

  1. It may be helpful to add a section on strengths/limitations near the end of the manuscript. Although the authors reference the strengths and limitations of the existing GBM literature, explicit acknowledgement of challenges in GBM epidemiology (e.g., short survival, cognitive changes that make case-control enrollment difficult, low incidence making prospective studies difficult, etc.) may be helpful to provide context for the overall findings. This would also provide a helpful area to describe possible future directions for GBM epidemiology (e.g., a greater inclusion of biomarker specific analysis of risk factors and incidence).

 RESPONSE:  Thank you for this suggestion. We added a section on limitations.

MINOR COMMENTS:

  1. It is a bit strange to begin the review stating that you will be including papers that do not include only the disease in question. While I agree that GBM-specific literature is limited and it is reasonable to include the glioma literature, this should be mentioned later in the introduction or discussion, rather than right at the outset.

 RESPONSE:  Thank you for this suggestion. We changed it.

  1. “Most malignant” (introduction, line 42), is a strange phrase. Also present in the Conclusion. Tumors are either malignant or not. Higher grade may be a better descriptor.

 RESPONSE:  Thank you for this suggestion. We changed  “most malignant” to “higher grade”.

  1. Figure 1 is excellent, Figures 2 and 3 should be improved. These are copied straight from Excel with very small font that makes interpretation difficult.

RESPONSE:  Thank you for this suggestion. We improved the figures 2 and 3.  

  1. Because Table 3 is present, Table 1 probably does not need to include incidence by gender, and can include overall only.

RESPONSE:  Thank you for this suggestion.

  1. There are other references available for antihistamines and glioma, including McCarthy et al. 2011 and an additional paper by Scheurer et al. (2008), among others.

RESPONSE:  Thank you for this suggestion. We added another reference for antihistamines and glioma (for example McCarthy et al. 2011, Scheurer et al 2008, Schlehofer et al., 1999, Schoemaker et al., 2006).

  1. There are other references available for statins and glioma, including Cote et al. (2019), Rendon et al. (2022), which show no significant association.

RESPONSE:  Thank you for this suggestion. We added reference about statins and glioma (for example Cote at al,. 2009, Rendon et al, 2022).

  1. The section on cannabinoids is quite long considering nearly all of the evidence for these is laboratory rather than population based

RESPONSE:  Thank you for this suggestion. We changed this section.

  1. I would argue that evidence for racial/ethnic disparities and glioma risk is quite strong. This section is also incorrectly titled “Ethnicity” when it should be “Race/Ethnicity”. Same comment for the title of Table 4.

RESPONSE:  Thank you for this suggestion. We changed “Ethnicity” to “Race/Ethnicity”.

  1. Section 6.8 is titled nutritional factors but does not discuss much of the rather rich literature on nutritional intake and glioma risk. For example, Kuan et al. (2019) provide an excellent study of nutrition and glioma showing no significant associations. Other studies have examined nutrients (e.g., flavonoids, vitamins) and glioma risk with some interesting findings.

RESPONSE:  Thank you for this suggestion. We added it.

  1. It would be reasonable to cite the original study by Stupp et al. in NEJM (2005) when discussing the Stupp regimen for GBM.

RESPONSE:  Thank you for this suggestion. We changed it.

Yours faithfully,

Anna Cybulska

Reviewer 2 Report

The authors have very well compiled the review entitled Epidemiology of Glioblastoma Multiforme-Literature review.GBM and glioma tumors are central nervous system tumors and need updated information.

The classification of GBM is described well and as well as the molecular markers for the genetic pathogenesis of GBM like ATRX,TERT,TP53,B-RAF,GATA4,EGFR etc are described in detail.

The survival and prognostic factors are listed as incidence,age,survival,urban/rural socioeconomic status provide insight into the prognosis of GBM.

A number of important risk factors are enumerated as Tobacco smoking and nitrosamines,ethnicity,ionizing radiation,head injury,obesity,growth metal etc which give in depth understaning of the underlying factors.

Treatment of GBM is also discussed.Overall an informative and focussed review on GBM.

The text in the figure # 2 could be enalrged.

Author Response

10th May 2022

Dear Reviewer,

We take the liberty to thank you for the insightful and careful evaluation of our article entitled “Epidemiology of Glioblastoma Multiforme - Literature Review ” by Szymon Grochans , Anna Maria Cybulska, Donata SimiÅ„ska, Jan Korbecki, Klaudyna Kojder, Dariusz Chlubek and Irena Baranowska-Bosiacka.

The comments helped us to improve the quality of the manuscript. We considered all comments and recommendations and responded to questions.

The correction throughout the manuscript were marked in yellow.

Our responses to the reviews are attached below.

Response to Reviewers

Comments and Suggestions for Authors

The authors have very well compiled the review entitled Epidemiology of Glioblastoma Multiforme-Literature review.GBM and glioma tumors are central nervous system tumors and need updated information.

The classification of GBM is described well and as well as the molecular markers for the genetic pathogenesis of GBM like ATRX,TERT,TP53,B-RAF,GATA4,EGFR etc are described in detail.

The survival and prognostic factors are listed as incidence,age,survival,urban/rural socioeconomic status provide insight into the prognosis of GBM.

A number of important risk factors are enumerated as Tobacco smoking and nitrosamines,ethnicity,ionizing radiation,head injury,obesity,growth metal etc which give in depth understaning of the underlying factors.

Treatment of GBM is also discussed.Overall an informative and focussed review on GBM.

The text in the figure # 2 could be enalrged.

RESPONSE:  Thank you for this suggestion.  We improved the figures 2.

Yours faithfully,

Anna Cybulska